# Breast Tumor Characterization Using [^18^F]FDG-PET/CT Imaging Combined with Data Preprocessing and Radiomics

**DOI:** 10.3390/cancers13061249

**Published:** 2021-03-12

**Authors:** Denis Krajnc, Laszlo Papp, Thomas S. Nakuz, Heinrich F. Magometschnigg, Marko Grahovac, Clemens P. Spielvogel, Boglarka Ecsedi, Zsuzsanna Bago-Horvath, Alexander Haug, Georgios Karanikas, Thomas Beyer, Marcus Hacker, Thomas H. Helbich, Katja Pinker

**Affiliations:** 1QIMP Team, Center for Medical Physics and Biomedical Engineering, Medical University of Vienna, 1090 Vienna, Austria; denis.krajnc@meduniwien.ac.at (D.K.); laszlo.papp@meduniwien.ac.at (L.P.); ecsedi.bogi@gmail.com (B.E.); 2Division of Nuclear Medicine, Department of Biomedical Imaging and Image-Guided Therapy, Medical University of Vienna, 1090 Vienna, Austria; thomas.nakuz@meduniwien.ac.at (T.S.N.); marko.grahovac@meduniwien.ac.at (M.G.); clemens.spielvogel@meduniwien.ac.at (C.P.S.); alexander.haug@meduniwien.ac.at (A.H.); georgios.karanikas@meduniwien.ac.at (G.K.); marcus.hacker@meduniwien.ac.at (M.H.); 3Division of Molecular and Gender Imaging, Department of Biomedical Imaging and Image-Guided Therapy, Medical University of Vienna, 1090 Vienna, Austria; heinrich.magometschnigg@meduniwien.ac.at (H.F.M.); thomas.helbich@meduniwien.ac.at (T.H.H.); katja.pinker@meduniwien.ac.at or pinkerdk@mskcc.org (K.P.); 4Christian Doppler Laboratory for Applied Metabolomics, Medical University of Vienna, 1090 Vienna, Austria; 5Department of Pathology, Medical University of Vienna, 1090 Vienna, Austria; zsuzsanna.horvath@meduniwien.ac.at; 6Memorial Sloan Kettering Cancer Center, Breast Imaging Service, Department of Radiology, New York, NY 10065, USA

**Keywords:** breast cancer, radiomics, machine learning, PET/CT, data pre-processing, triple negative

## Abstract

**Simple Summary:**

Breast cancer is the second most common diagnosed malignancy in women worldwide. In this study, we examine the feasibility of breast tumor characterization based on [^18^F]FDG-PET/CT images using machine learning (ML) approaches in combination with data-preprocessing techniques. ML prediction models for breast cancer detection and the identification of breast cancer receptor status, proliferation rate, and molecular subtypes were established and evaluated. Furthermore, the importance of most repeatable features was investigated. Results displayed high performance of malignant/benign tumor differentiation and triple negative tumor subtype ML models. We observed high repeatability of radiomic features for both high performing predictive models.

**Abstract:**

*Background*: This study investigated the performance of ensemble learning holomic models for the detection of breast cancer, receptor status, proliferation rate, and molecular subtypes from [^18^F]FDG-PET/CT images with and without incorporating data pre-processing algorithms. Additionally, machine learning (ML) models were compared with conventional data analysis using standard uptake value lesion classification. *Methods*: A cohort of 170 patients with 173 breast cancer tumors (132 malignant, 38 benign) was examined with [^18^F]FDG-PET/CT. Breast tumors were segmented and radiomic features were extracted following the imaging biomarker standardization initiative (IBSI) guidelines combined with optimized feature extraction. Ensemble learning including five supervised ML algorithms was utilized in a 100-fold Monte Carlo (MC) cross-validation scheme. Data pre-processing methods were incorporated prior to machine learning, including outlier and borderline noisy sample detection, feature selection, and class imbalance correction. Feature importance in each model was assessed by calculating feature occurrence by the R-squared method across MC folds. *Results*: Cross validation demonstrated high performance of the cancer detection model (80% sensitivity, 78% specificity, 80% accuracy, 0.81 area under the curve (AUC)), and of the triple negative tumor identification model (85% sensitivity, 78% specificity, 82% accuracy, 0.82 AUC). The individual receptor status and luminal A/B subtype models yielded low performance (0.46–0.68 AUC). SUV_max_ model yielded 0.76 AUC in cancer detection and 0.70 AUC in predicting triple negative subtype. *Conclusions*: Predictive models based on [^18^F]FDG-PET/CT images in combination with advanced data pre-processing steps aid in breast cancer diagnosis and in ML-based prediction of the aggressive triple negative breast cancer subtype.

## 1. Introduction

Breast cancer is the most common cancer in females, with over two million cases per year [1]. Among patients with a suspicious imaging abnormality at screening, image guided biopsy is used to confirm breast cancer diagnosis [2,3]. In breast cancer treatment assessment of receptor status (estrogen (ER), progesterone (PR) and Her2-neu receptor (HER2)) by immunohistochemistry (IHC) from breast biopsy is used for tumor subtype classification. Breast cancer molecular subtypes as determined by IHC (Luminal A, Luminal B, Her2 positive and Triple Negative) guide treatment decisions [4]. Nonetheless, breast cancer subtyping from biopsy sampling has limitations as it is subject to sampling bias and cannot fully capture intra-tumor heterogeneity [5,6,7]. In addition, there is its inherently invasive nature.

^18^F-fluorodeoxyglucose positron emission tomography/computed tomography ([^18^F]FDG-PET/CT) is a sensitive hybrid imaging method for detecting distant metastases and lymph node metastases in breast cancer patients [8] and for assessing treatment response [9,10]. Recently, dedicated PET/CT breast imaging protocols have shown potential for the classification and initial staging of primary tumors [11,12,13]. Despite first promising results for the non-invasive characterization of breast tumors, conventional PET/CT image analysis, including the standardized uptake value (SUV), tumor-to-background ratio (TBR), and metabolic tumor volume, remains of limited use for the differentiation of benign and malignant breast tumors and for molecular subtyping of breast cancers [14]. Therefore, several studies have performed radiomic analysis to further the value of [^18^F]FDG-PET/CT in this context [15,16,17].

Radiomic analysis combined with machine learning (ML) has shown promise for characterizing tumor heterogeneity [18,19], assessing therapy response [20,21,22], and improving prognostic stratification of cancer patients [23,24]. However, the lack of repeatability for radiomic models has been noted as a major bottleneck for a clinical adoption [25,26]. The Imaging Biomarker Standardization Initiative (IBSI) [27] as well as optimized radiomics [15] and ComBat feature normalization [28] have been proposed as methodological considerations to support building quantitative radiomic models that can be translated reliably into the clinics. Radiomics combined with ML is prone to challenges originating from the characteristics of the input data itself, such as low sample count [29], imbalanced disease subgroups [30,31], high-dimensionality of data [32,33], and outliers [34,35]. To address these limitations, data preparation steps are necessary [36,37], yet data preparation approaches remain underrepresented in the field of hybrid imaging radiomics. Considering that breast cancer molecular subtypes are naturally imbalanced, with one subgroup of a given subtype, such as more aggressive triple negative (TN) or HER2 positive, being significantly underrepresented than hormone receptor subtypes (ER/PR positive) [20,21,22], we hypothesize that breast cancer in vivo prediction models benefit from data preparation approaches.

Therefore, the objectives of this study are: (a) to establish prediction models for breast cancer detection and the identification of breast cancer receptor status, proliferation rate, and molecular subtypes from [^18^F]FDG-PET/CT images with ML, (b) to investigate the effect of data pre-processing on breast tumor characterization ML models, and (c), to compare ML-based prediction models with conventional SUV-based approaches.

## 2. Materials and Methods

### 2.1. Patients

One hundred and seventy patients (median age, 57.6 years; range, 18–86 years) were examined with [^18^F]FDG-PET/CT imaging between 2009 and 2014 as part of a prospective study, which has been previously reported [11,13,38] and approved by the institutional review board of the Medical University of Vienna (EK 510-2009). Written informed consent was obtained from all patients prior to the imaging examinations. The inclusion criteria were as follows: age 18 years or older; and an abnormality at mammography or breast ultrasound (asymmetric density, architectural distortion, suspicious microcalcifications, or breast mass classified as Breast Imaging Reporting and Data System (BI-RADS category 0 or 4–5). Exclusion criteria included pregnancy, lactation, prior treatment (e.g., breast biopsy before PET/CT, neoadjuvant chemotherapy), or inadequate patient positioning resulting in considerably compressed or deformed imaging. For all patients, the following clinical information was recorded: height, weight, body mass index (BMI), and age. See Figure 1 for the study design of our analysis.

### 2.2. Histopathologic Analysis

Diagnosis was established by an experienced specialized breast pathologist (ZBH). All lesions were verified by image-guided needle biopsy or surgery. For all invasive breast cancers, histopathology results were reviewed for tumor subtype according to the World Health Organization (WHO) classification [39], and tumor stage and grade according to Elston and Ellis [40]. Breast cancer intrinsic subtype was determined by immunohistochemistry based on estrogen receptor (ER), progesterone receptor (PR), human epidermal growth receptor 2 (HER2) status, and Ki-67 expression according to current guidelines [41], and defined as luminal A (ER/PR positive, Ki67 < 15%), luminal B (ER/PR positive, HER2 negative, Ki-67 ≥ 15% or ER/PR positive, HER2 positive), HER2 positive (ER/PR negative, HER2 positive), or triple negative (TN, ER/PR negative, HER2 negative) [42,43]. Patients with equivocal HER2 status were evaluated using chromogenic in situ hybridization to detect gene amplification. Patients with amplified genes were considered HER2 positive and patients whose genes were not amplified were considered as HER2 negative. In terms of Ki-67 expression, patients with ≥15% proliferation were considered as positive, while patients with <15% proliferation were classified as negative. HER2 positive and TN breast cancers were considered more aggressive breast cancers with a worse prognosis than luminal A/B breast cancers.

### 2.3. PET/CT

[^18^F]FDG-PET/CT of the breast was performed with a dedicated breast imaging protocol using a combined whole-body PET/CT system (Biograph 64 TruePoint^®^; Siemens Healthineers, Erlangen, Germany) with a high-resolution PET and a 64-row detector CT system. Patients were required to fast for at least 5 h before receiving an intravenous bolus injection of 200–350 MBq [^18^F]FDG based on body weight with blood glucose level < 150 mg/dL (8.3 mmol/L). After an uptake time of 60 min, PET/CT imaging was performed over one PET bed position with the patient consistently in the prone position [11,13,38]. The low-dose CT scan without CT contrast administration was acquired for attenuation correction covering a region from the base of the skull to the upper abdomen. Then, the PET acquisition was performed over the same region with 5 min acquisition time per bed position. CT images were reconstructed with 2 mm slice thickness. PET images were reconstructed using the iterative TrueX algorithm (Siemens), which incorporated resolution recovery [44,45]. Four iterations per 21 subsets were used with a matrix size of 168 × 168, a transaxial field of view (FOV) of 605 mm (pixel size of 3.6 mm), and a section thickness of 5 mm.

### 2.4. Lesion Delineation

PET/CT images were delineated in the Hybrid 3D software (ver. 4.0.0., Hermes Medical Solutions, Stockholm, Sweden). PET-based SUV values were normalized by a cubic volume of interest (VOI) over the mediastinum to serve as background reference for TBR calculations [46]. Three-dimensional isocount-based lesion delineations were performed semi-automatically on the PET images by a nuclear medicine specialist, and then reviewed by two radiologists (Figure 2). Based on previously suggested minimum voxel count for radiomic analysis [47] the smallest analyzed lesion size was 1.56 cm^3^. Overall, 167 patients had one primary lesion delineated, while three patients had two delineated lesions, resulting in overall 173 lesions.

### 2.5. Feature Extraction

Patient demographics (age, height, weight, body mass index (BMI)), conventional SUV PET (SUV_mean_, SUV_max_, SUV_min,_ SUV_peak_ and SUV_TLG_) and radiomic PET/CT features were combined to form a holomics dataset [25,48]. In order to support reproducibility of our study, radiomic features with “strong” and “very strong” consensus were extracted following the IBSI guidelines [27], combined with optimized feature extraction principles [15] from the 173 lesions. For each lesion, 48 PET features, 50 CT features, and 14 fusion PET/CT features were extracted and merged with patient demographics and SUV features, resulting in 121 features per lesion. See Appendix A for the list of IBSI-conform radiomic features.

### 2.6. Feature Redundancy Reduction

Covariance matrix analysis [49] was performed across the 120 features where features with absolute Pearson correlation coefficient greater than 0.95 were considered as redundant, resulting in 77 features for further analysis.

### 2.7. Predictive Model Establishment

Mixed ensemble learning of five Random Forest (RF) algorithms with various hyperparameter values [50] was utilized for model establishment to minimize the effect of method bias and to increase the predictive performance (Appendix A). The final model decision was obtained by majority vote across the five model predictions. The ensemble model scheme was utilized to establish breast cancer detection (malignant vs. benign), ER, PR, HER2, Ki-67, triple negative, and luminal A/B predictive models.

### 2.8. Model Performance Estimation

Hundred-fold Monte Carlo (MC) cross-validation with a training-to-validation ratio of 90%–10% was utilized for each model [51]. To estimate the performance of the established models compared with random guesses, sham data analysis was performed by random label permutations as done previously [37,52]. Confusion matrix (CM) analyses were employed to estimate model performance including accuracy (ACC), sensitivity (SENS), specificity (SPEC), positive predictive value (PPV), negative predictive value (NPV), and area under the receiver operator characteristics curve (AUC) across the MC folds.

### 2.9. Estimating the Effect of Data Preparation

This study utilized data preparation methods prior to ML over the training dataset of each MC fold. These methods covered a range of preprocessing steps including outlier and borderline sample detection [34,53], feature ranking and selection [54,55], and class imbalance correction [56,57,58]. Feature ranking was performed by R-squared approach [59] where the 15 highest-ranking feature per MC fold were selected from the training set for ML analysis. Methods were utilized in a predefined order of steps (Appendix A). In order to estimate the effect of these methods on ML predictive performance, each model was established twice within the Monte Carlo cross-validation scheme: with and without data preparation.

### 2.10. Feature Importance Estimation

To estimate the feature importance per predictive model, the feature occurrences as selected by the R-squared ranking were calculated across the individual MC folds.

### 2.11. Conventional PET Correlation Analyses

Conventional PET correlation analyses were performed for each patient subgroup according to malignant/benign tumor status, receptor status, proliferation rate, and molecular subtype. SUV_mean_, SUV_max_, SUV_min,_ SUV_peak_ and SUV_TLG_ PET-based features were analyzed by using the ANOVA *p*-value test method (Microsoft Excel 2016 software) with significance threshold of *p* < 0.05.

## 3. Results

### 3.1. Patients

Our cohort demonstrated highly imbalanced disease subgroups (Table 1). Out of 170 patients, 132 patients had a malignant breast tumor (78%) and 38 patients had a benign tumor (22%); 11 patients were classified as triple negative (6%), 22 as HER2 positive (13%), and 14 as luminal A (9%) vs. 81 as luminal B (81%). Furthermore, 88 patients were ER positive (52%), 78 were PR positive (46%), and 73 had a high number of Ki-67 positive cells (43%).

### 3.2. Model Performance Estimation

#### 3.2.1. Breast Cancer Detection

The model for differentiation of benign and malignant breast tumors/breast cancer detection with data preparation yielded 80% sensitivity, 78% specificity, 80% accuracy and 0.81 AUC, compared to the same model without data preparation (80% sensitivity, 59% specificity, 69% accuracy and 0.71 AUC). See Figure 3 for the performance comparison of the breast cancer detection models.

#### 3.2.2. Breast Cancer Subtyping

The highest cross-validation performance was achieved with the molecular subtyping ML model for triple negative breast cancer with data preparation which yielded 85% sensitivity, 78% specificity, 82% accuracy and 0.82 AUC. In contrast, the same model without data preparation yielded 59% sensitivity, 94% specificity, 75% accuracy and 0.76 AUC. See Figure 4 for the performance comparison of the triple negative models.

Data preparation did not impact ML model performance for the prediction of individual receptor status and proliferation rate (0.46–0.68 AUC).

Table 2 summarizes the Monte Carlo cross-validation performance of all ensemble predictive models with and without data preparation. Predictive performance of all models over sham data yielded 0.47–0.59 AUC (Appendix A).

### 3.3. Feature Importance Estimation

#### 3.3.1. Breast Cancer Detection

In the cancer detection predictive model, nine out of ten most relevant features for breast cancer detection originated from PET images. Features with highest occurrence number (n = 100) were five PET gray level co-occurrence matrix (GLCM) features (sum entropy, energy, difference entropy, information correlation 1, dissimilarity), two PET histogram features (skewness, uniformity), PET neighborhood grey tone difference matrix (NGTDM) contrast and SUV_max_ feature. High occurrence was also observed in PET GLCM joint maximum (n = 92), SUV_mean_ (n = 90) and patient demographics age feature (n = 85). See Figure 5 for all selected features in the breast cancer detection ML model.

#### 3.3.2. Breast Cancer Subtyping

In the triple negative predictive model eight out of ten most relevant features originated from PET images. Features with highest occurrence number (n = 100) were four PET GLCM features (contrast, difference entropy, dissimilarity, sum average), two PET NGTDM features (contrast, strength), PET histogram kurtosis, PET intensity range, PET + CT fusion cluster shade and SUV_mean_ feature. High occurrence was also observed in PET GLCM sum entropy (n = 93), SUV_max_ (n = 93), PET GLSZM large zone high grey level emphasis (n = 89), PET GLCM cluster prominence (n = 88) and PET histogram skewness feature (n = 81). See Figure 6 for all selected features in the triple negative ML model.

### 3.4. Conventional PET Correlation Analysis

Appendix A summarizes SUV correlation metrics for malignant-vs-benign breast tumors (SUV_max_
*p* = 0.0002) as well as malignant tumors stratified by receptor status and molecular triple negative subtype (SUV_max_
*p* = 0.000001). Significant differences in SUV_max_ distributions were present in ER (SUV_max_
*p* = 0.00016), PR (SUV_max_
*p* = 0.003), and Ki-67 (SUV_max_
*p* = 0.003) subgroups. In contrast, HER2 (SUV_max_
*p* = 0.54) and luminal A/B subgroups (SUV_max_
*p* = 0.81) demonstrated low correlation.

The highest performance of the SUV models was demonstrated by SUV_max_ in cancer detection (0.76 AUC) and predicting triple negative subtype (0.70 AUC). See Figure 7 for comparison of AUC performance of SUV_max_ and holomics-based predictive models in cancer detection and triple negative subtype. See Appendix A for comparison of holomics-based and SUV-based ML predictive performance across all models.

## 4. Discussion

This study investigated the performance of ML predictive models based on [^18^F]FDG-PET/CT ML analysis of 173 breast tumors in 170 patients with and without data preparation. Our study shows that data pre-processing contributes to model performance of the breast cancer detection ML model (80% vs. 69% accuracy, 0.81 vs. 0.77 AUC) and the aggressive triple-negative breast cancer subtype ML model (82% vs. 75% accuracy, 0.82 vs. 0.76 AUC). Nonetheless, our findings regarding the molecular subtype ML models also imply that data pre-processing alone does not warrant performance improvement, unless there is already an identifiable pattern in the imbalanced subgroups. The low performance observed in our molecular subtype ML models is in agreement with radiomics studies investigating the predictability of molecular subtypes in breast cancer with dynamic contrast-enhanced magnetic resonance imaging (MRI) and diffusion-weighted imaging [60,61,62]. We consider that the low performance of these models is due to the fact that compared with individual receptors, molecular subtypes such as the triple negative subtype are determined by the information from all receptors and carry distinct radiomics signatures.

Feature importance estimation analysis in our two highest performing models (breast cancer detection and triple negative, both with data preparation) revealed that PET is the most important information source to establish these models. Specifically, 16 out of 18 prominent features and 18 out of 20 prominent features selected across MC folds were from PET in the two models respectively (Figure 5 and Figure 6). Furthermore, only two and three of PET features were conventional SUV parameters respectively. The prominent role of SUV_max_ was identifiable in both models (n = 100 for breast cancer detection and n = 93 for triple negative) implying, that radiomics and conventional SUV metrics in combination can maximize the predictive performance of these models instead of building on only one of these feature sets. These findings are in alignment with recent studies investigating the importance of PET radiomic and SUV parameters in characterizing tumors in vivo [37,50]. Radiomic PET feature types represented a wide range in both models, where most prominent features were extracted from the neighborhood gray tone difference (NGTDM) and gray level co-occurrence (GLCM) matrices. These matrices are both designed to describe heterogeneity characteristics of lesions [27]. As an example, the NGTDM contrast feature which was identified as high-ranking (n = 100) in both models, reflects on spatial intensity changes in between neighboring voxels.

The low importance of CT features in both models needs to be interpreted with caution. While CT may not represent heterogeneity patterns on its own, it contributes to PET attenuation correction as part of the PET/CT hybrid scanner [63], therefore, any prominent role identified in PET is inherently influenced by the presence of CT as well. Furthermore, the GLCM cluster shade PET/CT fusion feature was identified with high importance (n = 100) in the breast cancer detection. This feature implies that the co-occurrence pattern of spatially-overlapping PET and CT voxels can contribute to differentiate malignant and benign breast lesions.

Patient age was the only demographics feature identified as highly important (n = 85) in the breast cancer detection model [64], while it was present with negligible importance (n = 11) in the triple negative predictive model. To date, no studies of cancer detection or triple negative subtype prediction based on PET/CT radiomics in compliance with IBSI has been performed, hence the comparison of our findings on feature repeatability to other studies remains of interest.

To date, studies that have analyzed radiomic features based on [^18^F]FDG-PET/CT in patients with breast cancer have focused on building models to predict pathological complete response to neoadjuvant chemotherapy or to differentiate breast carcinoma from breast lymphoma [20,22,65]. Antunovic et al. reported AUCs of 0.70–0.73 across all predictive models [22]; Li et al. reported AUCs of 0.72 and 0.73, respectively, without and with patient age incorporated [20]; and Ou et al. reported AUCs of 0.81 and 0.76 for PET and CT models, respectively [65]. Huang et al. [66] reported mean AUCs of 0.75 and 0.68 for one-year and two-year recurrence-free survival, respectively, from using PET/MRI-based models. The sample size of the patient cohort in all these studies ranged from 44–113 patients, which is lower compared to that used in our work (n = 170). In addition, none of the above studies utilized data preprocessing approaches and they also did not build on mixed ensemble learning to minimize method selection bias in their relatively small patient cohorts.

To date, data pre-processing steps have been rarely discussed in PET-based radiomic studies. Zhou et al. [67] implemented class imbalance correction in a breast cancer cohort of 55 patients in their MRI-based radiomics study; specifically, the minority subclass was corrected using the Synthetic Minority Oversampling Technique (SMOTE). They reported AUCs of 0.81–0.87 across six different predictive MRI-based models to predict response to neoadjuvant chemotherapy [67]. Cysouw et al. [37] performed imbalance correction applying the SMOTE algorithm, and in addition they utilized principal component analysis to reduce the high number of features while retaining 95% of the observed variance to characterize prostate cancer in [^18^F]DCFPyL PET. Xie et al. [36] investigated class imbalance solutions in a cohort of head and neck cancer patients in their [^18^F]FDG-PET/CT-based radiomics study, by testing various resampling techniques for generating minority subclass samples and for cleaning noisy and redundant data. They reported performance increase of 0.32 (AUC) with applying data resampling techniques.

Our study differed in several aspects to the aforementioned studies. First, Zhou et al. and Cysouw et al. did not consider the presence of noisy/borderline data samples which may decrease the overall performance of the established models [68,69,70]. Second, none of the prior studies handled outliers in their training datasets.

In our study, conventional PET-based correlation analysis showed that SUV_max_, SUV_mean_, and SUV_min_ were significantly different between malignant and benign tumors, which is in agreement with prior publications [11,71,72]. For predicting individual breast cancer receptor status and proliferation rate, conventional PET correlation analysis showed that SUV_max_ and SUV_mean_ were significantly different according to ER and PR status, and according to Ki-67 protein expression. For predicting molecular subtype, significance differences in standard SUV metrics were found only for the triple negative subtype (SUV_max_ and SUV_mean_, *p* < 0.001 for both) while for the HER2 and luminal A/B subtype, standard SUV metrics showed no significant differences. Although there were significant differences in SUV metrics for ER and PR expression and the triple negative subtype, SUV-based models resulted in poor predictive performance.

Compared with the radiomics ML models, the SUV-based models had a lower AUC performance for differentiating between malignant and benign tumors (0.76 AUC vs. 0.81 AUC) as well as triple negative subtypes (0.70 AUC vs. 0.82 AUC). Performance difference is even more expressed in other confusion matrix analytics metrics, where we observed lower accuracy (ACC) performance of SUV-based models in cancer detection (56% vs. 80%) as well as in triple negative subtype prediction (74% vs. 82%).

Our study had limitations: our analysis was based on data from a single center only. Nevertheless, we extracted highly-repeatable radiomic features from our data as of the IBSI-standard together with optimized radiomic parameter sets [15,27]. In addition, we utilized Monte Carlo cross-validation scheme in combination with ensemble learning to minimize the effect of selection bias in both our data and ML methods. Last, we performed sham data analysis by random label permutation to estimate the performance compared to random guess.

Considering the high repeatability of our identified high-ranking features and the high reproducibility nature of RF classifiers [73], we consider that our findings could be reproduced by other centers building on our methodological approaches. The results of our study indicate that future radiomic studies can benefit from data pre-processing steps before conducting ML analyses especially if the given disease subgroups are highly imbalanced.

Of note, this study did not aim to investigate solid vs. invasive tumor sensitivities independently, rather, the effect of data preparation on overall ML performance which analyzes all types of tumors. Nevertheless, lesion delineation in our study was PET-driven. PET has high sensitivity, but at the same time it is prone to partial volume effects which naturally results in overestimating lesion volumes. Therefore, we assume that boundary regions of invasive tumors were not underrepresented in our analysis.

## 5. Conclusions

The diagnostic accuracy of [^18^F]FDG-PET/CT of breast cancer detection and prediction of the aggressive triple negative molecular subtype of breast cancer improved following the use of advanced data pre-processing in radiomic models. Radiomics analysis of [^18^F]FDG-PET/CT aid the differentiation of benign and malignant tumors in patients that cannot be assessed sufficiently with conventional breast imaging and who are not candidates for MRI. Results indicate that aggressive triple negative breast cancers that often require intensified or presurgical treatment carry distinct radiomics signatures and can be separated from less aggressive subtypes. However, radiomics analysis of [^18^F]FDG-PET/CT is limited in value for the prediction of individual receptor status and proliferation rate.

## Figures and Tables

**Figure 1 cancers-13-01249-f001:**
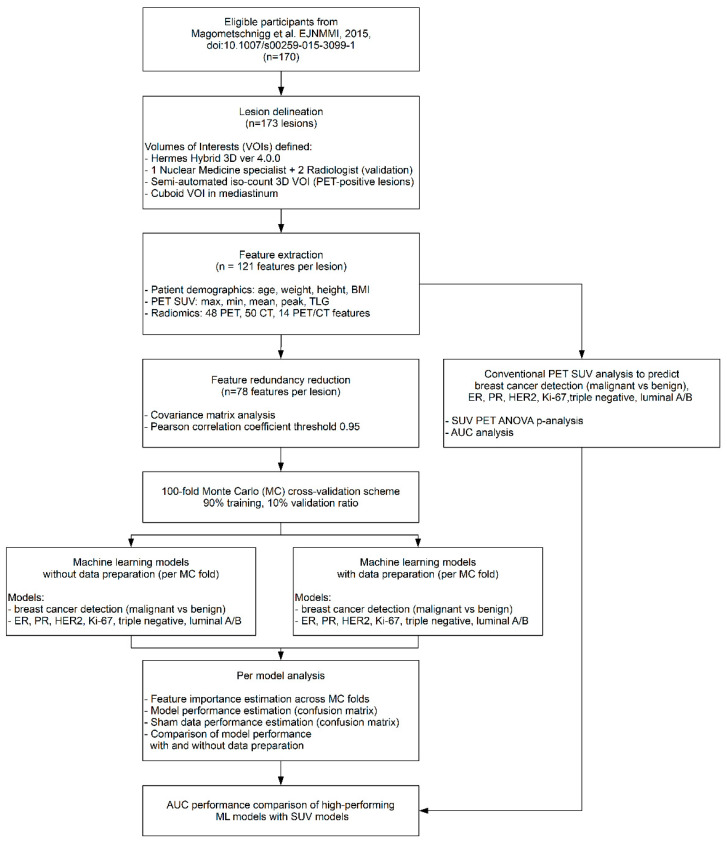
The analysis workflow of the collected dataset. Prospective study conducted between 2009 and 2014, approved by the institutional review board provided data records for 170 patients. [^18^F]FDG-PET/CT of the breast was performed with a dedicated breast imaging protocol using a combined whole-body PET/CT system. 173 lesions were delineated and extracted following the imaging biomarker standardization initiative (IBSI) guidelines combined with optimized feature extraction principles. Feature redundancy reduction was performed resulting in 77 features. Monte Carlo cross validation was utilized to generate 100 training vs. validation folds. Pre-processing steps were performed over training data. Ensemble learning scheme was utilized to establish predictive models. All machine learning models underwent confusion matrix analytics, sham data analysis, and Area Under the Receiver Operator Characteristics Curve (AUC) analysis across MC folds and the conventional PET SUV analysis. VOI = Volume of Interest; BMI = Body Mass Index; ER = Estrogen; PR = Progesterone; HER2 = Human Epidermal Growth Receptor 2; PET–Positron Emission Tomography; CT–Computed Tomography.

**Figure 2 cancers-13-01249-f002:**
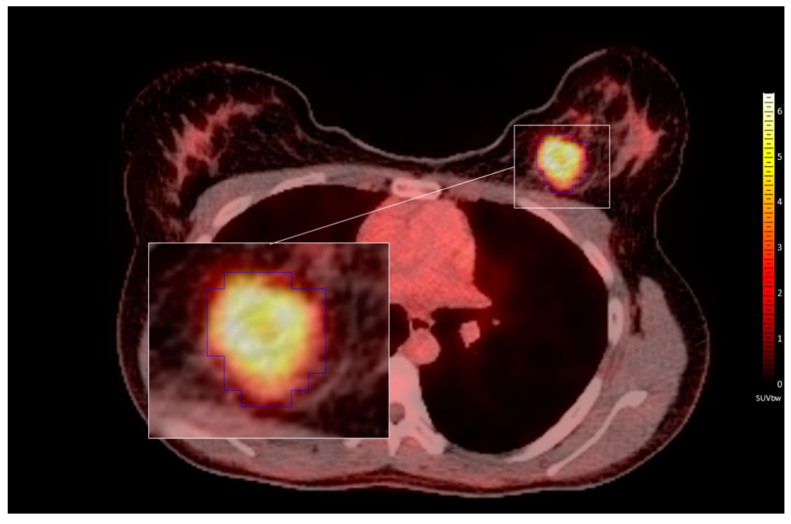
18F-fluorodeoxyglucose positron emission tomography/computed tomography ([^18^F]FDG-PET/CT) view of a breast cancer patient with semi-automatically delineated volume of interest (VOI) in the PET image. Windowing: hot iron palette with SUV body weight (SUV_bw_) of 6.5 for PET and range of −100 to 200 Hounsfield units (HU) for CT. The patient underwent imaging procedure in prone position and view is shown following the radiological convention.

**Figure 3 cancers-13-01249-f003:**
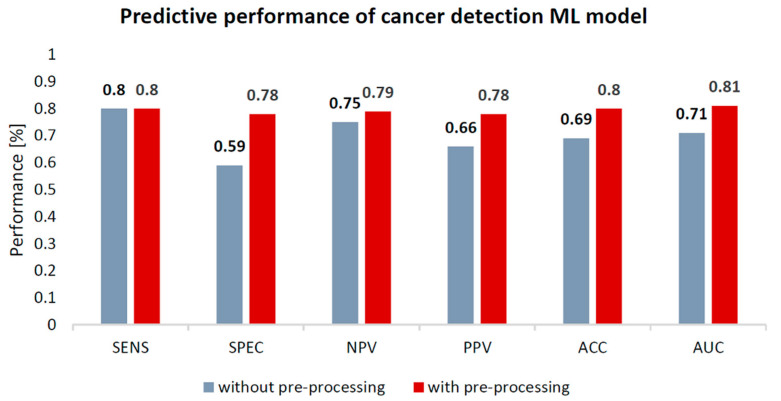
Performance comparison of breast cancer detection machine learning (ML) predictive models, with and without data pre-processing. ACC = Accuracy; SENS = Sensitivity; SPEC = Specificity; NPV = Negative Predictive Value; PPV = Positive Predictive Value. Performance is expressed in percentages (%).

**Figure 4 cancers-13-01249-f004:**
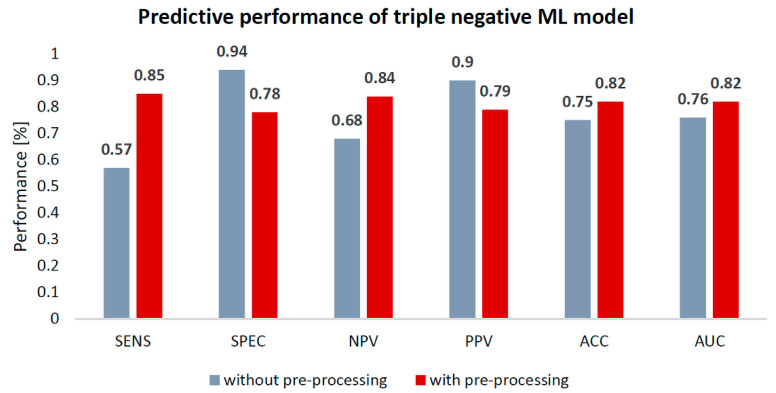
Performance comparison of triple negative subtype machine learning (ML) predictive models, with and without data pre-processing. ACC = Accuracy; SENS = Sensitivity; SPEC = Specificity; NPV = Negative Predictive Value; PPV = Positive Predictive Value. Performance is expressed in percentages (%).

**Figure 5 cancers-13-01249-f005:**
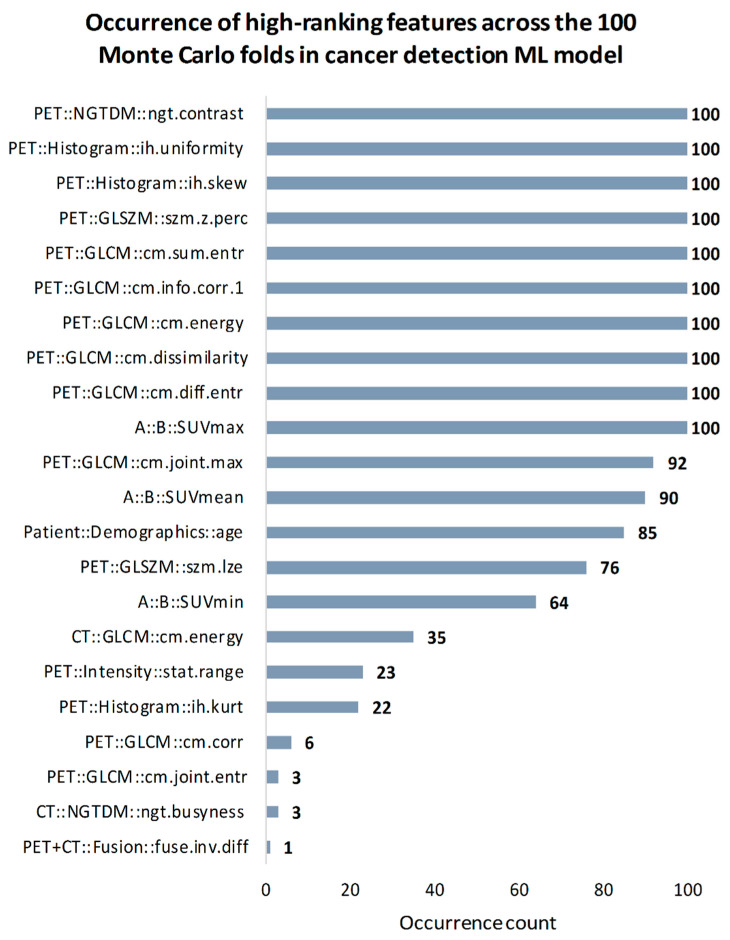
Occurrence of high-ranking features across the 100 Monte Carlo folds in cancer detection predictive model. NGTDM = neighborhood grey tone difference matrix; GLSZM = gray level size zone matrix; GLCM = gray level co-occurrence matrix; SUV_max_ = maximum standard uptake value; SUV_mean_ = mean standard uptake value; SUV_min_ = minimal standard uptake value; skew = skewness; z.perc = zone percentage; entr = entropy; info.corr.1 = information correlation 1; joint.max = joint maximum; lze = large zone emphasis; kurt = kurtosis; corr, correlation; joint.entr = joint entropy; inv.diff = inversed difference.

**Figure 6 cancers-13-01249-f006:**
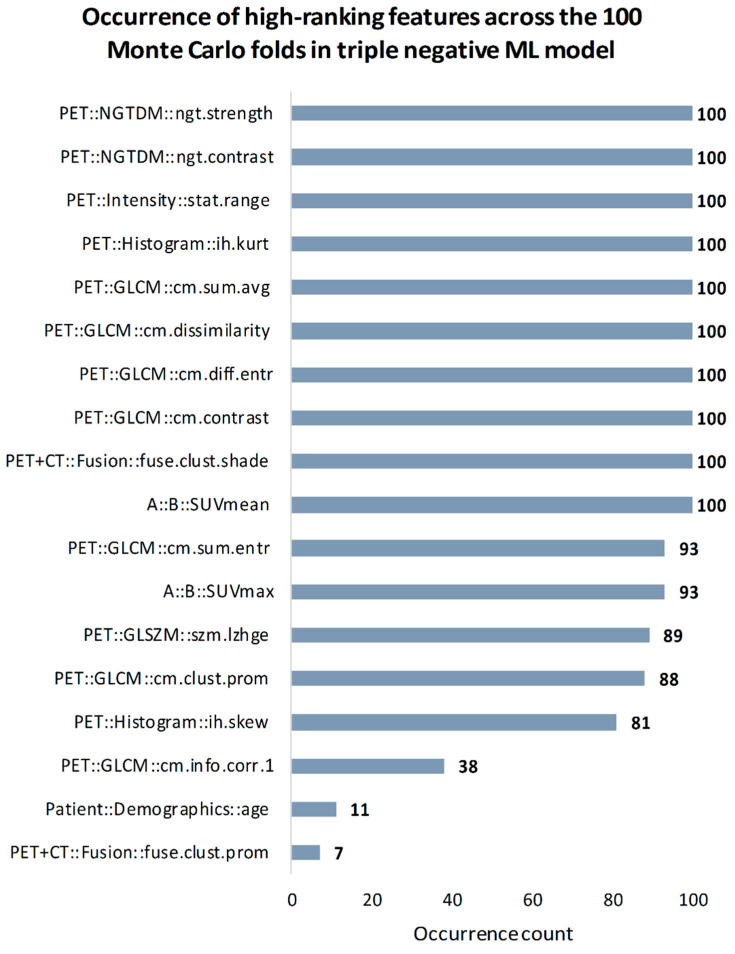
Occurrence of high-ranking features across the 100 Monte Carlo folds in triple negative predictive model. NGTDM = neighborhood grey tone difference matrix; GLCM = gray level co-occurrence matrix; GLSZM = gray level size zone matrix; SUV_max_ = maximum standard uptake value; SUV_mean_ = mean standard uptake value; kurt = kurtosis; sum.avg = sum average; diff.entr = difference entropy; clust.shade = cluster shade; sum.entr = sum entropy; lzhge = large zone high grey level emphasis; clust.prom = cluster prominence; skew = skewness; info.corr.1 = information correlation 1.

**Figure 7 cancers-13-01249-f007:**
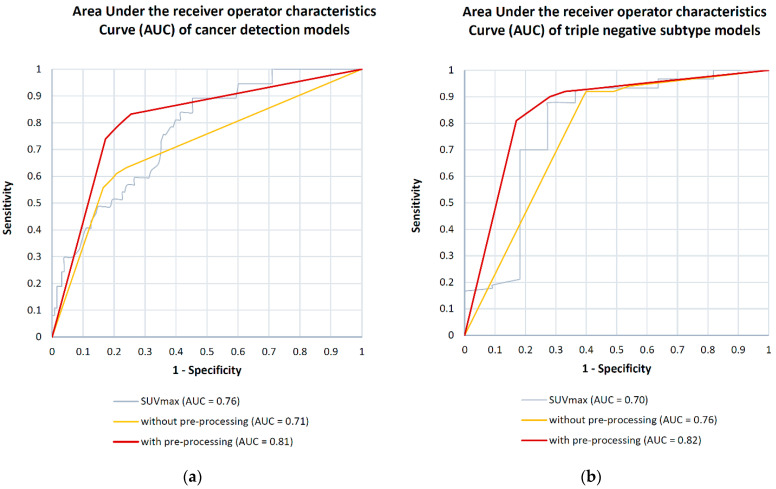
Comparison of area under the receiver operator characteristics curve (AUC) performance of maximum standard uptake value (SUV_max_) and holomics-based ensemble models with and without data pre-processing for (**a**) breast cancer detection (**b**) triple negative subtype.

**Table 1 cancers-13-01249-t001:** Patient cohort characteristics for malignancy, estrogen (ER), progesterone (PR), human epidermal growth receptor 2 (HER2), Ki-67 protein expression, triple negative, and luminal A/B status. NA = Not Available.

Patient Characteristics (n = 170)	Value
Age (years), median (IQR)	57.6 (18–86)
Lesion volume (cm^3^), median (IQR)	12.8 (6.2–26.9)
Malignancy	n (%)
Malignant	132 (78)
Benign	38 (22)
Estrogen (ER)	n (%)
−	17 (10)
+	88 (52)
NA	65 (38)
Progesterone (PR)	n (%)
−	27 (16)
+	78 (46)
NA	65 (38)
Ki-67	n (%)
−	26 (15)
+	73 (43)
NA	71 (42)
HER2	n (%)
−	84 (49)
+	22 (13)
NA	64 (38)
Triple negative	n (%)
Yes	11 (6)
No	95 (56)
NA	64 (38)
Luminal A/B	n (%)
A	14 (8)
B	81 (48)
NA	75 (44)

**Table 2 cancers-13-01249-t002:** Monte Carlo cross-validation performance of all ensemble predictive models with and without data preparation. Confusion matrix values are expressed in percentages (%). AUC is expressed in ratio.

Model	Data Preprocessing	SENS	SPEC	NPV	PPV	ACC	AUC
**ER**	No	83	40	70	58	62	0.63
	Yes	82	56**↑**	78**↑**	65**↑**	69**↑**	0.68**↑**
**PR**	No	74	36	58	54	55	0.56
	Yes	78**↑**	35	61**↑**	54	56**↑**	0.55
**Ki-67**	No	68	39	55	53	53	0.63
	Yes	65	45**↑**	56**↑**	54**↑**	55**↑**	0.65**↑**
**HER2**	No	17	84	50	51	50	0.46
	Yes	17	84	50	51	50	0.46
**Luminal A/B**	No	17	87	51	57	52	0.62
	Yes	16	89**↑**	51	59**↑**	53**↑**	0.52
**Triple negative**	No	57	94	68	90	75	0.76
	Yes	85**↑**	78	84**↑**	79	82**↑**	0.82**↑**
**Breast Cancer Detection (Malignant vs. Benign)**	No	80	59	75	66	69	0.71
	Yes	80	78**↑**	79**↑**	78**↑**	80**↑**	0.81**↑**

ACC = Accuracy, AUC = Area under the receiver operator characteristic curve, SENS = Sensitivity, SPEC = Specificity, NPV = Negative Predictive Value, PPV = Positive Predictive Value, ER = Estrogen, HER2 = Human Epidermal Growth Receptor 2, PR = Progesterone. Sign ^↑^ indicates performance increase in pre-processed training datasets compared to original datasets.

## Data Availability

Available upon reasonable request.

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
