# Peer review of "Breast Tumor Characterization Using [18F]FDG-PET/CT Imaging Combined with Data Preprocessing and Radiomics"

_cancers, 2021, doi:10.3390/cancers13061249_

Round 1
Reviewer 1 Report
The submitted manuscript has organized clinical observation for radiological diagnosis of cancer. The metadata and data optimization for better clinical diagnosis is interesting in the manuscript. Most of part looks pretty OK. However, there is little information needed for clarification. For example-
- The introduction section should have minimum scientific jargon to focus the finding in the results section. The authors may write the introduction to describe the keywords of the manuscript.
- Since it is a clinical study, the institutional review board's name and approved authentication id should mention in the experimental section.
- In the results and discussions section, the authors need to focus on this study's objectives are (a) to establish prediction ... (b) ….
- The conclusion section should have sufficient remarks with discussion for the general reader.
Author Response
Reviewer 1
The submitted manuscript has organized clinical observation for radiological diagnosis of cancer. The metadata and data optimization for better clinical diagnosis is interesting in the manuscript. Most of part looks pretty OK. However, there is little information needed for clarification. For example-
Q1. The introduction section should have minimum scientific jargon to focus the finding in the results section. The authors may write the introduction to describe the keywords of the manuscript.
A1. Thank you for the comment. We have revised the introduction as suggested and trust it is more focused.
Q2. Since it is a clinical study, the institutional review board's name and approved authentication id should mention in the experimental section.
A2. We have added the information per reviewer’s request. (Line 101)
“One hundred and seventy patients (median age, 57.6 years; range, 18–86 years) were examined with [18F]FDG-PET/CT imaging between 2009 and 2014 as part of a prospective study, which has been previously reported in different contexts [11,13,38] and approved by the institutional review board of the Medical University of Vienna (EK 510-2009).”
Q3. In the results and discussions section, the authors need to focus on this study's objectives are (a) to establish prediction ... (b) ….
A.3 Thank you for the comment. We have revised the results as suggested. (Lines: 251-253; 264-266; 294-296; 314)
Q4. The conclusion section should have sufficient remarks with discussion for the general reader.
A4. Thank you for the comment. We have revised the conclusion as suggested. (Lines 475-484)
“We have demonstrated that the diagnostic accuracy of [18F]FDG-PET/CT of breast cancer detection and prediction of the aggressive triple negative molecular subtype of breast cancer is improved when advanced data pre-processing is employed in radiomic models. Radiomics analysis of [18F]FDG-PET/CT aid the differentiation of benign and malignant tumors in patients that cannot be assessed sufficiently with conventional breast imaging and who are not candidates for MRI. Results indicate that aggressive triple negative breast cancers that often require intensified or presurgical treatment carry distinct radiomics signatures and can be separated from less aggressive subtypes. For prediction of individual receptor status and proliferation rate radiomics analysis of [18F]FDG-PET/CT demonstrate limitations..”
Reviewer 2 Report
Could authors discuss the smallest size of detectable tumor?
Also, what is the relative sensitivity for solid tumor well-defined boundary versus invasive tumor with ill-defined boundary?
Author Response
Reviewer 2
Comments and Suggestions for Authors
Q1: Could authors discuss the smallest size of detectable tumor?
A1: Thank you for the comment. We added the information about the smallest size of detectable tumor as suggested. (Lines: 170-171)
“Based on previously suggested minimum voxel count for radiomic analysis [47] the smallest analyzed lesion size was 1.56 cm3”
Q1: what is the relative sensitivity for solid tumor well-defined boundary versus invasive tumor with ill-defined boundary?
A1: Thank you for this interesting question! We did not aim to investigate solid vs invasive tumor sensitivities independently in our study, rather, the effect of data preparation on overall machine learning performance which analyzes all types of tumors. Nevertheless, PET has high sensitivity, but at the same time it is prone to partial volume effects. Since delineation was PET-driven, we assume that boundary regions of invasive tumors were not underrepresented in our analysis. We revised the discussion section and included this information. (Lines: 468-473)
“Of note, this study did not aim to investigate solid vs invasive tumor sensitivities in-dependently, rather, the effect of data preparation on overall ML performance which analyzes all types of tumors. Nevertheless, lesion delineation in our study was PET-driven. PET has high sensitivity, but at the same time it is prone to partial volume effects which naturally results in overestimating lesion volumes. Therefore, we assume that boundary regions of invasive tumors were not underrepresented in our analysis.”
Reviewer 3 Report
No major issues.
Author Response
Thank you.
Reviewer 4 Report
The authors of this manuscript present predictive models based on PET-CT images in combination with advanced data pre-processing in breast cancer and machine learning methods. The manuscript is well written and presented. The manuscript also includes an appropriate limitations section in the discussion. This manuscript adds to the growing literature regarding predictive models and machine learning and the authors should be congratulated for their work.
Author Response
Thank you.
Round 2
Reviewer 1 Report
The editor can consider accepting the review manuscript.